# Natural-Origin Betaine Surfactants as Promising Components for the Stabilization of Lipid Carriers

**DOI:** 10.3390/ijms25020955

**Published:** 2024-01-12

**Authors:** Agata Pucek-Kaczmarek, Dominika Celary, Urszula Bazylińska

**Affiliations:** Laboratory of Nanocolloids and Disperse Systems, Department of Physical and Quantum Chemistry, Faculty of Chemistry, Wroclaw University of Science and Technology, Wybrzeze Wyspianskiego 27, 50-370 Wroclaw, Poland; dominika.celary99@gmail.com

**Keywords:** solid lipid nanoparticles, nanostructured lipid carriers, cocamidopropyl betaine, coco betaine, green chemistry, colloidal stability

## Abstract

In the present work, we demonstrate studies involving the influence of the formulation composition on the physicochemical properties of nanocarriers: solid lipid nanoparticles (SLNs) and nanostructured lipid carriers (NLCs). Novel lipid-origin platforms were prepared using two “green” betaine-based surfactants, cocamidopropyl betaine (ROKAmina K30) and coco betaine (ROKAmina K30B), in combination with three different solid lipids, cetyl palmitate (CRODAMOL CP), trimyristin (Dynasan 114), and tristearin (Dynasan 118). Extensive optimization studies included the selection of the most appropriate lipid and surfactant concentration for effective SLN and NLC stabilization. The control parameters involving the hydrodynamic diameters of the obtained nanocarriers along with the size distribution (polydispersity index) were determined by dynamic light scattering (DLS), while shape and morphology were evaluated by atomic force microscopy (AFM) and transmission electron microscopy (TEM). Electrophoretic light scattering (ELS) and turbidimetric method (backscattering profiles) were used to assess colloidal stability. The studied results revealed that both betaine-stabilized SLN and NLC formulations containing CRODAMOL CP as lipid matrix are the most monodisperse and colloidally stable regardless of the other components and their concentrations used, indicating them as the most promising candidates for drug delivery nanosystems with a diverse range of potential uses.

## 1. Introduction

Green chemistry, based on environmentally safe ingredients and low-cost production methods, is a promising approach to designing more sustainable and safe commercial products. Currently, due to the growing trend of ecological solutions in the pharmaceutical and cosmetic industry, a search for new, advanced active substance delivery systems can be observed [1]. Lipid-origin colloidal formulations, composed of lipid (solid at room temperature as well as at body temperature), surfactant(s), and water, cover a wide range of advantages, such as high biocompatibility and low toxicity of nanocarriers, increased stability as well as capacity of a controlled release of the encapsulated drug and possibility of sterilization/large-scale production [2,3]. Solid lipid nanoparticles (SLNs), a primary class of lipid formulations, are made only of solid lipids, creating a crystalline and organized structure. In the second generation nanostructured lipid carriers (NLCs), a portion of a solid lipid is replaced by a liquid lipid (oil); due to this, the lipid matrix changes to a more imperfect one, which reduces the order of the structure, resulting in an increase in the space available for the encapsulated drug [4]. Drug delivery platforms based on solid lipids have allowed some disadvantages of the other nanocarrier types to be overcome. When taking into account the low kinetic stability in the case of emulsions, expensive and sensitive components for the production of liposomes, low drug encapsulation and loading efficiency of micelles, and use of organic solvents in the case of polymeric particles, lipid nanoparticles seem to be effective, environmentally friendly and economical. Additionally, the use of a solid lipid as a matrix enables the protection of thermolabile active substances [5,6,7]. Among the ways of application of colloidal systems based on solid lipids extensively documented in the literature are cosmetic [5,8,9], food [4,6,10], and, above all, in the pharmaceutical industry (i.e., drug delivery) [7,11,12]. In recent years, lipid formulations have enabled the effective encapsulation of active substances successfully delivered intravenously [13,14,15], orally [16,17,18], transdermally [19,20,21], and also directly into the eye [22,23,24] or nose [25,26,27].

Colloidal stability is one of the most important parameters in designing formulations for delivering active agents. The attainment of durable physical and chemical stability of lipid nanoparticles is also crucial from the point of view of prolonged storage of the final product. Designing stable lipid nanocarriers requires the appropriate choice of surfactants playing the role of emulsifiers, which reduce the surface tension and aid in particle dispersion in any applicable formulation [28]. In this context, the selection of the appropriate surfactant and the emulsifier concentration seems to be essential. Furthermore, the composition of the formulation has a necessary influence on safety during administration in the body and penetrability into the cell [29]. Accordingly, betaines, used in the present contribution as stabilizing agents for lipid nanocarriers, are a class of compounds characterized by quaternized nitrogen. In alkyl-betaine (coco betaine), one of the methyl groups in the betaine structure (i.e., N,N,N-trimethylglycine) was replaced by a linear alkyl chain. Additionally, in alkylamido-betaine (cocamidopropyl betaine), an amide group is a linker between the hydrophobic alkyl chain and the hydrophilic moiety. In contrast to quaternary ammonium salts, betaines do not have a “mobile” anion and maintain their positive molecular charge as well as cationic character in acidic and alkaline media. Therefore, it is correct to classify them as cationic emulsifiers, not as amphoteric surface active agents [30]. Nevertheless, unlike cationic surfactants, their unique semi-synthetic structure of green origin based on coconut oil provides them unique properties typical for biosurfactants, i.e., biodegradability, biocompatibility, non-toxicity, and gentleness of use, with a simultaneous high stabilizing and solubilizing ability. Therefore, both of these compounds have recently been successfully used to stabilize nanoemulsion and other micellar formulations, mostly for transdermal applications [31,32].

Taking the above into account, the purpose of this work (Figure 1) was to design, engineer, and optimize the composition of novel lipid nanoparticles (i.e., SLNs and NLCs) involving the recent leanings for accessing “green” materials, i.e., natural-origin betaine-based surfactants, cocamidopropyl betaine, and coco betaine (commercial products: ROKAmina K30 and ROKAmina K30B, respectively), and using low-energy consuming, i.e., sustain ultrasonic-emulsification technique. This approach is highly effective and efficient with readily available laboratory equipment. It is also a “green” method, in which the use of organic solvents (which are usually toxic) can be avoided [11]. The influence of the formulation composition (i.e., type of the lipid and surfactant and the substrate concentration) on the final dispersion properties (i.e., particle size; polydispersity index; and zeta potential, shape, and morphology, as well as colloidal stability) was carried out to develop the potential practical application. Our studies present the first and original approach involving mild coconut oil-origin surfactants in the efficient stabilization of novel SLNs and NLCs nanocarriers with cosmetic and pharmaceutical impact.

## 2. Results and Discussion

In the present study, solid lipid nanoparticles (SLNs) and nanostructured lipid carriers (NLCs) were fabricated by the ultrasonic-emulsification method. As the lipid matrix materials, three solid lipids were chosen: cetyl palmitate (CRODAMOL CP), trimyristin (Dynasan 114), and tristearin (Dynasan 118). The proposed lipids were selected from the top ten most popular as less expensive, green, and easily available lipid phases with high potential in SLN and NLC formulation. Furthermore, they are classified as Generally Recognized as Safe (GRAS) and produced with Good Manufacturing Practice (GMP) [33]. For stabilization, two betaine-based surfactants, cocamidopropyl betaine (ROKAmina K30) and coco betaine (ROKAmina K30B), were used.

The impact of composition parameters (amount of lipid phase: 1, 2, or 3% *w*/*w*) and surfactant content in the water phase (1 or 2% *w*/*w*) on the nanocarrier properties was determined. The synthesis parameters (ultrasonication time, used amplitude, and number of cycles) were kept constant. Table 1 and Table 2 summarize the details of the obtained sample composition. The influence of the ingredients on size (hydrodynamic diameter, D_H_) and polydispersity index (PdI) of the obtained nanocarriers were evaluated at production time (t = 0 days) and during a storage period of 50 days using dynamic light scattering (DLS). The most promising systems were assessed by electrophoretic light scattering (ζ-potential) and turbidimetric method (backscattering profiles) as well as by atomic force microscopy (AFM) and transmission electron microscopy (TEM).

### 2.1. Solid Lipid Nanoparticles

The produced SLNs stabilized by cocamidopropyl betaine (ROKAmina K30) presented values of hydrodynamic diameters (Figure 1a) in the range of 150–430 nm at the preparation time (0 days). Only two formulations (SLN_5_K30 and SLN_6_K30) were not stable and solidified after 50 days (no results within 50 days on the bar graph). Most of the obtained systems were characterized by a relatively low polydispersity index (PdI < 0.3), which confirms good homogeneity (Figure 1b). The highest PdI was presented by carriers obtained at a lipid concentration of 3%. In other cases, changes in the size of the carriers and their polydispersity were not significant. It is worth emphasizing that with increasing concentration of the lipid phase (and constant concentration of cocamidopropyl betaine in the dispersion), the size of nanocarriers increases, and the homogeneity of the obtained systems decreases, which results in a higher PdI. The presented results are in good correlation with the other lipid nanocarriers reported in the literature [34], indicating a significant effect on lipid concentration, size, and PdI of obtained formulations.

The type of lipid used as matrix also has a high impact on the physicochemical characteristics of the system—carriers based on cetyl palmitate (CRODAMOL CP) have the smallest size (samples SLN_13_K30—SLN_18_K30), which certainly gives them the greatest possibilities due to subsequent application. Formulations based on this lipid also proved to be the most stable (the smallest changes in size 50 days after preparation) and were thus selected for subsequent zeta potential measurements. Nanoparticles obtained using Dynasan 118 as a lipid matrix (SLN_7_K30—SLN_12_K30) were characterized by the largest size (above 200 nm). However, the type of lipid used had no significant effect on the polydispersity index of the nanoparticles.

The obtained SLNs stabilized by coco betaine (ROKAmina K30B) presented diameter values (Figure 2a) in the range of 80–230 nm at the preparation time (0 days). In some cases, the size of the carriers exceeded 1000 nm (SLN_3_K30B and SLN_9_K30B), or the formulations solidified quickly (SLN_11_K30B and SLN_12_K30B)—these samples are not shown in Figure 2. Some samples (SLN_4_K30B, SLN_5_K30B, and SLN_6_K30B) were not stable and solidified after 50 days (no results within 50 days on the bar graph). This phenomenon is known in the literature and called “gelling” and is usually caused by a concentration of the lipid phase that is too high [35]. Most of the produced SLNs were characterized by a relatively low polydispersity index (PdI < 0.3), which confirms good homogeneity (Figure 2b). In the case of ROKAmina K30-stabilized SLNs, samples with ROKAmina K30B as a surfactant and CRODAMOL CP as lipid matrix (samples from SLN_13_K30B to SLN_18_K30B) proved to be the most stable and homogeneous, regardless of the component concentrations used. Also, in the case of this surfactant, carriers with CRODAMOL CP as lipid matrix were the most homogeneous and stable in terms of size and polydispersity (samples SLN_13_K30—SLN_18_K30) (samples SLN_13_K30B—SLN_18_K30B). The size of these lipid systems below 200 nm and the narrow size distribution also meet the criteria of the most desirable nanocarriers of bioactive compounds for medical and pharmaceutical applications administrated intravenously [36].

Since CRODAMOL CP resulted in being the most effective lipid in the synthesis of SLNs stabilized by betaine-based surfactant, samples 13–18 were selected for electrophoretic light scattering (ELS) measurements to determine to determine zeta potentials (ζ-potential). The results for both surfactants (cocamidopropyl betaine and coco betaine) are presented in Figure 3a and Figure 3b, respectively. The zeta potential of the formulations was determined in the original dispersion media [37].

The absolute value of ζ-potential for suspensions considered highly stable should be about 30 mV [38]. The obtained results show that the nanocarriers with a lower zeta potential were obtained using higher surfactant concentrations. This is consistent with other data regarding the effect of surfactant concentration on the zeta potential of lipid carriers given in the literature data [39,40].

The observed high colloidal stability of the obtained nanoparticles may result from the mutual electrostatic repulsion of the surfactants, i.e., between the ammonium groups (which are quaternary) and the carboxylate groups [41]. It is also possible that the high stability of the carriers can be caused not only by the electrostatic repulsion but also by some steric stabilization of the formulation that may occur since both betaine-based surfactants possess the alkyl chains long enough to make this mechanism effective [42]. Comparing systems stabilized with various surfactants—formulations with coco betaine have slightly lower zeta potentials than those with cocamidopropyl betaine, but consequently, both systems confirmed good colloidal stability, which was also proved in the backscattering profiles (Figure 4).

The results of backscattering profiles (BS) allow for the detection of sample instability and, therefore, the determination of the kinetic stability. The dispersions were rated using the Turbiscan Lab Expert optical analyzer based on multiple light scattering technology. The X-axis shows the level of the colloidal sample in the glass vial (the height of the sample, expressed in mm). The slight changes (after 50 days) in BS are only observed for formulation stabilized by coco betaine (Figure 4b). These carriers may be less stable, which was also confirmed by electrophoretic light scattering (lower zeta potentials, Figure 3). Rapid destabilization is often described by large gaps between the curves (deviations above 10%), and overlapping of individual curves suggests a slow destabilization process and, thus, high stability of the analyzed formulation [43,44].

The optimized SLNs with the most favorable physicochemical features (i.e., contained CRODAMOL CP as lipid matrix (sample systems SLN_17_K30 and SLN_17_K30B)) were also selected for the morphology imaging provided by two powerful and quick microscopic techniques: transmission electron microscopy (TEM) and atomic force microscopy (AFM). From Figure 5, we can observe spherical objects with a size of about 200 nm without any aggregation and deformation. Moreover, the obtained images do not show major morphological differences depending on the type of surfactant used (Figure 5a,b). Interestingly, in the case of TEM, we can observe a clear, tidy-packed lipid core and a darker surfactant shell. The structure is typical for internal turbulence of the SLN as described in the literature data [2,3,4,5]; however, it is not often observed in microscopic images. It is also worth noting that the size and polydispersity of the systems obtained are in agreement with the measurements made using DLS.

### 2.2. Nanostructured Lipid Carriers

The NLCs stabilized by cocamidopropyl betaine (ROKAmina K30) presented size values (Figure 6a) in the range of 170–320 nm at the preparation time (0 days). All prepared formulations were stable for at least 50 days. Also, all of the obtained systems were characterized by a relatively low polydispersity index (PdI < 0.3), which confirms good homogeneity (Figure 6b). The changes in the size of the carriers and their polydispersity were not significant after 50 days. Similarly to the SLNs, nanocarriers with the addition of liquid oil based on Dynasan 118 as solid lipid (samples NLC_5_K30 to NLC_8_K30 and NLC_17_K30 to NLC_20_K30) resulted in being the largest in size. Replacing the solid lipid with CRODAMOL CP (samples NLC_9_K30 to NLC_12_K30 and NLC_21_K30 to NLC_24_K30) resulted in a reduction in the diameter of the nanoparticles and the PdI.

An increase in the concentration of liquid oil in the carrier matrix has a positive effect on its parameters. All formulations with a solid/liquid ratio of 6:4 have lower diameters and better PdI than samples with a ratio of 8:2. The obtained results are in good agreement with other data concerning the design and development of lipid nanoparticles reported in the literature [45,46].

The obtained NLCs stabilized by coco betaine (ROKAmina K30B) presented diameter values (Figure 7a) in the range of 40–170 nm on the day of synthesis (0 days). All obtained formulations were stable (50 days) and characterized by a low polydispersity index (PdI < 0.3), which confirms good homogeneity (Figure 7b). The changes in the size and polydispersity were not significant after storage time. It is worth emphasizing that the addition of liquid oil in the matrix reduces the size of the carriers and decreases their polydispersity (compared to SLNs). The effect of liquid oil concentration in the lipid matrix is the same as in the case of carriers stabilized by cocamidopropyl betaine—adding more liquid oil decreases the particle size of NLCs.

The zeta potential of the NLC formulations with CRODAMOL CP (as the most homogeneous in size and polydispersity) is presented in Figure 8. The results indicate that the use of cocamidopropyl betaine (Figure 8a) as a stabilizing agent effect in a lower zeta potential value (between −20 and −30 mV), which may translate into better formulation stability than with coco betaine (Figure 8b). This relationship was also confirmed in the backscattering profiles presented in Figure 9, where the scattering lines for freshly obtained samples are almost identical to those stored for 50 days. This means that no destabilization processes, such as aggregation or sedimentation, appear in the optimized NLCs sample.

The optimized NLCs with the most favorable physicochemical features (sample system 24_K30 and system 24_K30B) were also selected for the size and shape imaging provided TEM and AFM to compare the morphology of these systems with SLNs analogs contained CRODAMOL CP as lipid matrix. It can be seen from Figure 10 that these objects are different from the images shown in Figure 5. The NLCs are less regular in shape, semi-spherical objects with a size of above 100 nm, as the internal structure of the NLCs is different from that of the SLN, and the degree of order is caused by the presence of liquid oils in the structure of the matrix of nanostructured carriers. As it has been proven by many previous researchers, the mixture of solid and liquid fats leads to a less ordered internal structure of the particle [2,3]. It is worth noting that, similarly to SLNs, no significant differences were observed for systems stabilized by different surfactants, both for TEM and AFM images. Moreover, the results obtained are also in good correlation with those obtained using DLS.

## 3. Materials and Methods

### 3.1. Materials

Solid lipids for nanoparticle production were supplied as follows: CRODAMOL CP (cetyl palmitate; CP) from CRODA, and Dynasan 114 (trimyristin; D114) and Dynasan 118 (tristearin; D118) from CREMER OLEO. Liquid oil, Miglyol 812N (medium-chain triglyceride oil; M812), was purchased from CREMER OLEO. Betaine-based surfactants ROKAmina K30 (cocamidopropyl betaine) and ROKAmina K30B (coco betaine) were gifted by PCC ROKITA (structures are shown in Figure 1). Distilled water was used for all experiments.

### 3.2. Preparation of Solid Lipid Nanoparticles and Nanostructured Lipid Carriers

Lipid nanocarriers were prepared using an ultrasonic-emulsification method described in our previous publication [47]. For SLN formulation, the lipid phase containing the appropriate compound (D144, D118, or CP) of known mass (corresponding to 1, 2, or 3% by mass of the total formulation) was heated up above the melting point of the lipid. For NLC formulation, in the lipid phase, a portion of the solid lipid (20 or 40%) was replaced by liquid oil (M812). Surfactants (ROKAmina K30 or ROKAmina K30B) were dispersed in distilled water (5 mL), heated to the same temperature, and added dropwise to the melted lipid (followed by magnetic stirring). The formed pre-emulsion was subjected to ultrasonic treatment (power 100 W) using an ultrasonic Lab Homogenizer UP100H (Hielscher Ultrasonics GmbH, Teltow, Germany). The process was performed for constant period (8 min), with 100% power output (amplitude) in a continuous mode. The samples were allowed to cool down at room temperature and then stored at 4 °C for further analysis. The quantitative composition details of the various formulations (SLNs and NLCs) are presented in Table 1 and Table 2, respectively.

### 3.3. Characterization Methods

#### 3.3.1. Particle Size and Polydispersity Index by Dynamic Light Scattering

The measurements of size and polydispersity index of the obtained formulations were performed using dynamic light scattering (DLS) by a Malvern Zetasizer Nano ZS (Malvern Instruments, Worcestershire, UK). Measurements were carried out with the detection angle of 173° using optically homogeneous square polystyrene cells. DLS gives values of the hydrodynamic diameter (D_H_) of nanocarriers as an intensity-weighted mean diameter of the bulk population, while the polydispersity index (PdI) is a measure of the width of the particle size distribution. Each value was derived from the average of three consecutive instrument runs (with at least 10 measurements). All the measurements were conducted at a temperature of 25 °C.

#### 3.3.2. Surface Charge by Electrophoretic Light Scattering

The charge of particles (i.e., zeta potential, denoted as ζ-potential) of the obtained formulations was determined using the electrophoretic method by a Malvern Zetasizer Nano ZS apparatus (Malvern Instruments, Worcestershire, UK). A field intensity of 20 V/cm was used. The results are presented as the mean of three measurements (each with at least 20 runs). All the measurements were performed at a temperature of 25 °C.

#### 3.3.3. Morphological Characterization by Atomic Force Microscopy and Transmission Electron Microscopy

The shape and morphology of the SLNs and NLCs were estimated by atomic force microscopy (AFM) using a Veeco NanoScope Dimension V AFM (Plainview, New York, NY, USA) with a tube scanner (type RT ESP Veeco). The scanning speed of 0.5 Hz was used. During the measurements, a low-resonance-frequency pyramidal silicon cantilever resonating at 250–331 kHz was employed with a constant force of 20–80 N/m. Before observations, formulations were diluted, placed on a fresh mica surface (by dipping it in solution), and allowed to adsorb for about 24 h. Then, the surfaces were rinsed with water and dried at room temperature.

The morphology was also investigated by transmission electron microscopy (TEM) using an FEI Tecnai G2 20 X-TWIN microscope (FEI, Hillsboro, OR, USA). A drop of the diluted sample was placed on a perforated, carbon-film-coated copper grid and was left to dry at room temperature before the examination.

#### 3.3.4. Kinetic Stability by Multiple Light Scattering

The physical stability of the studied dispersions was performed using the TurbiScanLab Expert (Formulaction SA, Toulouse, France). The analysis was performed at a temperature of 25 °C in cylindrical glass cells. Measurements of backscattering (BS) were carried out with pulsed near-infrared light (λ = 880 nm), collecting BS profiles as a function of sample height. The results were analyzed by the instrument software Turbisoft (version 2.2.0.82).

## 4. Conclusions

Our contribution revealed novel “green” lipid nanosystems, i.e., solid lipid nanoparticles (SLNs) and nanostuctured lipid carriers (NLCs) formulated by cocamidopropyl betaine (ROKAmina K30) and coco betaine (ROKAmina K30B) in combination with three different solid lipids: cetyl palmitate (CRODAMOL CP), trimyristin (Dynasan 114), and tristearin (Dynasan 118). The key control parameters, including size, polydispersity, zeta potential, backscattering profiles, and nanoparticle morphology, indicated the most favorable SLN and NLC systems for further cosmetic, pharmaceutical, or biomedical applications. In both cases, the nanoparticles containing CRODAMOL CP as a lipid matrix proved to be the most colloidally stable and homogeneous (with size below 200 nm and PdI < 0.3), regardless of the other components and the concentrations used. Consequently, these optimized SLNs and NLCs were indicated for further experiments involving encapsulation processes for the delivery of many hydrophobic active agents with high biological impact. Furthermore, these experiments for lipid origin nanocarriers stabilized by “green” betaine-based surfactants should also involve in vitro drug release profiles in different environmental and biological stability in various conditions, biocompatibility analysis, evaluation of the cytotoxic effect on cell lines, as well as assessment of the in vivo intracellular distribution of nanoparticles, to study the quality, efficacy, and safety profile of selected systems as drug-delivery nanoplatforms.

## Data Availability

All the data supporting the findings of this study are available within the article. Raw data are available from the corresponding authors on a reasonable request.

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
