# Peer review of "Natural-Origin Betaine Surfactants as Promising Components for the Stabilization of Lipid Carriers"

_ijms, 2024, doi:10.3390/ijms25020955_

Round 1

Reviewer 1 Report

Comments and Suggestions for Authors

Targeted delivery systems based on lipid particles are not new, but they do not lose their relevance. The transition from synthetic components to natural ones is the main trend of modern chemistry. The creation of lipid nanoparticles based on natural components is a logical development of these delivery methods.

There are a number of technical comments regarding the presented work.

1) The abbreviations from table 1 do not match the legends of the samples in Figures 1-3 and 6-8. I recommend that the authors correct the table in accordance with the figures. It's easier than correcting so many illustrations.

2) Based on the values of the zeta potential, which reaches 30 mV in only 3 samples, I have a question about the stability of the particles. In the classical concept of colloidal particles with a charge, particles with a potential of more than 30 mV are considered stable. The authors do not directly explain the reason for the stability of their particles.

3) the composition of the particles and their packaging have also not been assessed.

4) All AFM images do not have a scale; it must be provided.

5) The scale on TEM images is difficult to read, it is necessary to make it more contrasty for perception.

6) There are also voaros for TEM. A typical burnout of organic matter under the beam is observed. To estimate sizes using this method, metal spraying is required to stabilize the particles. In addition, it is necessary to provide a histogram of particle size distribution.

Reviewer 2 Report

Comments and Suggestions for Authors

The current manuscript is an interesting experimental article on the development of novel lipid nanocarriers using betaine-derived surfactants. Many relevant experiments were performed, and it seems to be overall well done. Nevertheless, some alterations are necessary before acceptance for publication:

- The abstract should be modified to include some of the study’s specific results and conclusions, since it only includes introduction, objective and some methodology information;

- In the introduction section, more should be said about nanosystems as therapeutical options, and the choice for NLC and SLN instead of other nanosystem types should be justified: what are their characteristics, advantages and disadvantages, especially compared to other nanocarrier types (nanoemulsions, polymeric nanoparticles, micelles, etc.)?;

- The chosen formulation components, other that betaine-derived surfactants, should be justified: why these ones and not others? And what are the specific functions of each of them in the formulations?;

- Figure 1 quality (resolution) should be improved; additionally, statistical tests should be performed to evaluate whether the size and PDI differences between formulations are statistically significant; the same should be done in all similar figures, regarding size, PDI, zeta potential, etc.;

- Future experiments should be mentioned, namely the evaluation of drug release using model hydrophobic compounds, and possible cellular and/or in vivo studies;

- An abbreviation list should be added.

Round 2

Reviewer 1 Report

Comments and Suggestions for Authors

The authors took into account most of the comments. The article may be published

Regarding the last question, namely “In addition, it is necessary to provide a histogram of the particle size distribution.”

Particle size distribution is very important. And for the development of your project, namely the processing of microphotographs, I recommend using the free software imageJ (https://imagej.net/ij/download.html). With its help, you can select statistics for the image and build a distribution of histograms by size in Excel or originPro.